# Retrovirus-Derived *RTL9* Plays an Important Role in Innate Antifungal Immunity in the Eutherian Brain

**DOI:** 10.3390/ijms241914884

**Published:** 2023-10-04

**Authors:** Fumitoshi Ishino, Johbu Itoh, Masahito Irie, Ayumi Matsuzawa, Mie Naruse, Toru Suzuki, Yuichi Hiraoka, Tomoko Kaneko-Ishino

**Affiliations:** 1Department of Epigenetics, Medical Research Institute (MRI), Tokyo Medical and Dental University (TMDU), Tokyo 113-8510, Japan; irie@dna-gib.com (M.I.); am@logomixgenomics.com (A.M.); mnaruse@ncc.go.jp (M.N.); 2Department of Pathology, School of Medicine, Tokai University, Isehara 259-1193, Japan; itohj@is.icc.u-tokai.ac.jp; 3Faculty of Nursing, School of Medicine, Tokai University, Isehara 259-1193, Japan; 4Department of Genomic Function and Diversity, Medical Research Institute (MRI), Tokyo Medical and Dental University (TMDU), Tokyo 113-8510, Japan; 5Laboratory of Genome Editing for Biomedical Research, Medical Research Institute (MRI), Tokyo Medical and Dental University (TMDU), Tokyo 113-8510, Japan; t-suzuki.lra@mri.tmd.ac.jp (T.S.); yhiraoka.aud@mri.tmd.ac.jp (Y.H.); 6Laboratory of Molecular Neuroscience, Medical Research Institute (MRI), Tokyo Medical and Dental University (TMDU), Tokyo 113-8510, Japan

**Keywords:** retrovirus-derived gene *RTL9*, fungi, brain innate immunity, microglia, zymosan, lysosome, eutherian-specific traits, development and evolution

## Abstract

Retrotransposon Gag-like (RTL) genes play a variety of essential and important roles in the eutherian placenta and brain. It has recently been demonstrated that *RTL5* and *RTL6* (also known as *sushi-ichi retrotransposon homolog 8* (*SIRH8*) and *SIRH3*) are microglial genes that play important roles in the brain’s innate immunity against viruses and bacteria through their removal of double-stranded RNA and lipopolysaccharide, respectively. In this work, we addressed the function of *RTL9* (also known as *SIRH10*). Using knock-in mice that produce RTL9-mCherry fusion protein, we examined RTL9 expression in the brain and its reaction to fungal zymosan. Here, we demonstrate that *RTL9* plays an important role, degrading zymosan in the brain. The RTL9 protein is localized in the microglial lysosomes where incorporated zymosan is digested. Furthermore, in *Rtl9* knockout mice expressing RTL9ΔC protein lacking the C-terminus retroviral GAG-like region, the zymosan degrading activity was lost. Thus, RTL9 is essentially engaged in this reaction, presumably via its GAG-like region. Together with our previous study, this result highlights the importance of three retrovirus-derived microglial RTL genes as eutherian-specific constituents of the current brain innate immune system: *RTL9*, *RTL5* and *RTL6*, responding to fungi, viruses and bacteria, respectively.

## 1. Introduction

Retrotransposon Gag-like (RTL, also known as sushi-ichi retrotransposon homolog (SIRH)) genes are eutherian-specific except for therian-specific *Paternally expressed 10* (*PEG10*) [1,2,3,4,5]; comparative genome analyses have provided convincing evidence that *PEG10* emerged in a therian common ancestor while others are conserved exclusively in eutherians [5,6,7], indicating that they are presumably derived from an extinct retrovirus similar to the sushi-ichi retrotransposon. This view is supported by the evidence that the gypsy retrotransposon to which the sushi-ichi retrotransposon belongs is an infectious retrovirus of *Drosophila melanogaster*, possessing an *ENV*-like gene in addition to *GAG* and *POL* [8,9].

Among the 11 RTL genes, *PEG10* [1,2], *RTL1* (also known as *PEG11*) [10,11] and *Leucine zipper down-regulated in cancer 1* (*LDOC1*, also known as *SIRH7* and *RTL7*) [12,13,14] play essential but different roles in the placenta while *RTL4* (also known as *SIRH11* and *Zinc finger CCHC domain-containing 16* (*ZCCHC16*)) plays an important role in controlling impulsivity in the brain [14,15] and is recognized as a causative gene in autism spectrum disorders [16]. Moreover, *RTL1* plays important roles in the muscle and brain and is considered to be one of the major genes responsible for Kagami–Ogata and Temple syndromes [11,14,17,18,19,20].

We have recently demonstrated that *RTL5* (also known as *SIRH8*) and *RTL6* (also known as *SIRH3*) are microglial genes playing important roles in the brain’s innate immunity against viruses and bacteria by their removal of double-stranded (ds)RNA and lipopolysaccharide (LPS), respectively [14,21]. Microglia are the primary immune cells in the brain, playing a central role in the innate immune responses to various pathogens via a variety of Toll-like receptors (TLRs) that induce inflammation via regulation of cytokine and interferon expression [22,23]. It is well known that the TLR system is widely conserved in the animal kingdom from worms to mammals [24], while *RTL5* and *RTL6* are apparently novel constituents of the eutherian innate immune system [14,21].

Innate antifungal immunity is another important survival mechanism of organisms in addition to those against bacteria and viruses. Over the past few decades, the incidence of invasive opportunistic fungal infections has increased substantially, and antifungal drug resistance has also increased [25,26,27,28,29]. Fungal diseases are associated with increased morbidity and mortality, particularly in immunocompromised individuals. Therefore, research focused on understanding the molecular and cellular basis of antifungal immunity has expanded dramatically in recent years [25,26,27,28,29,30,31]. Zymosan is frequently used to experimentally induce an inflammatory reaction, as it mimics fungal infection. It is a component of the yeast cell wall that consists of protein–carbohydrate complexes, such as glucans, mannans, proteins, chitins and glycolipids. It is recognized by several cell surface receptors, such as TLR2, TLR6 and Dectin-1, becomes incorporated into phagosomes and is ultimately degraded in the phagolysosomes present in macrophages/microglia [30,31,32,33].

Phagocytes, including macrophages/microglia, play a central role in antifungal immunity using a variety of oxidative and non-oxidative mechanisms that work synergistically to kill extracellular and internalized fungi by producing reactive oxygen intermediates and/or hydrogen peroxide, such as hypochlorous acid and hypoiodous acid in the former activity and antibacterial peptides and hydrolases in the latter [30,31]. Lysosomes are organelles in animal cells that act as degradation centers for a variety of biomolecules, including carbohydrates, proteins, lipids and nucleotides, with various hydrolytic enzymes functioning in an acidic environment (pH 4.5–5.0) [34,35].

*RTL9* (also known as *SIRH10* or *Retrotransposon GAG domain containing 1* (*RGAG1*)) is another RTL gene that is highly conserved in eutherians, but the biological function of which has remained undetermined. In this report, we show that the RTL9 protein localizes in microglial lysosomes and plays an essential role in the degradation of fungal zymosan. Thus, it turns out that eutherians came to possess at least three eutherian-specific pathways against fungi, viruses and bacteria as a result of the respective acquisition of *RTL9*, *RTL5* and *RTL6* in the innate immune system of the brain.

## 2. Results

### 2.1. RTL9 Is Conserved in Eutherians

*RTL9* encodes a large protein comprising 1347 amino acids (aa) and is evolutionarily conserved in eutherians (dN/dS < 1) (Table 1 and Appendix A), suggesting it confers certain evolutionary advantage(s).

Like the other RTL genes, *RTL9* exhibits homology to the sushi-ichi retrotransposon GAG, but is unique in that it further exhibits homology to two herpes virus-like sequences, the outer envelope glycoprotein BLLF1 (super family member pfam05109, the gray box in the figure corresponding to 280–594 aa) and the large tegument protein UL36 (super family member pha03247, the orange box corresponding to 787–1141 aa) (Figure 1a, Appendix A). The BLLF1 and UL36 proteins have several functional motifs and domains, but there are no such functional sequences present in RTL9 (Appendix A, see also the Section 3). More detailed homology analysis has indicated that sushi-ichi GAG in fact exhibits a high degree of homology with a widespread region, including the capsid domain (the blue box corresponding to 1169–1250 aa) (Figure 1a). One such region is the one corresponding to 1148–1367 aa, from just before the capsid region to the end of RTL9 (the minimal GAG-like region, the red dashed box in Figure 1a, identity 25.5%, similarity 43.1%) (Appendix A) with another being the region from 927 to 1367 aa (the maximal GAG-like region indicated by the red double-headed arrow in Figure 1a, identity 21.8%, similarity 35.7%) corresponding to the entire GAG sequence (Appendix A). Thus, it remains ambiguous whether the 927–1148 aa portion originated from UL36 or GAG. In any event, information from the protein sequence alone is not sufficient to infer the biological function of the RTL9 protein.

*Rtl9* mRNA expression was very weak, detected only in the brain, lung, skeletal muscle and uterus after 32 PCR cycles, and was almost absent from most tissues and organs at 19 weeks (19 w) in adult mice (Figure 1b). Similar results were reported for human *RTL9* (GTEx Portal site, searched as *RGAG1*): low expression in the brain, heart and ovary in addition to a relatively higher expression in the testis (Appendix A). Therefore, for the in vivo detection of RTL9, we generated an *Rtl9-mCherry* knock-in (KI) mouse that expresses the RTL9-mCherry (RTL9CmC) protein fused with a fluorescent mCherry protein after the C-terminus of endogenous RTL9 (Figure 1c, top and Appendix A). We also generated an *Rtl9ΔC* mouse (hereafter called the *Rtl9* KO mouse) that produces RTL9 with a truncated C-terminus (922–1367 aa, the maximal GAG-like region, Figure 1a and Appendix A) to elucidate the function of RTL9, especially of its C-terminal GAG-like region and potentially including the last portion of the UL36-like region (927–1145 aa).

### 2.2. RTL9 Localizes in Microglial Lysosomes

The RTL9CmC protein is highly expressed in microglial lysosomes in the neonatal brain. Using *Rtl9-CmCherry* KI mice, we detected the mCherry signal in the midbrain and around the thalamus in postnatal day 0–15 (P0–15) neonates in regions such as the superior colliculus (SC), cerebral aqueduct (AQ), periaqueductal gray (PAG) matter, posterior commissure (PC) and the commissures of the superior and inferior colliculus (CSC and CIC) (Figure 2a, left). The mCherry signal was specific to the *Rtl9-CmCherry* in the KI brain and almost negligible in the control WT brain (Figure 2a, right), confirming that the mCherry signal correctly reflects the localization of the RTL9CmC protein. Its expression profile remained almost the same until 3~4 weeks of age when the signal strength progressively diminished, and this trend continued up to the adult period.

On higher magnification, the signals were detected in a number of small granules (approximately 1 μm diameter or less) that accumulated near the nucleus in round cells (Figure 2b). We confirmed that these RTL9-expressing cells are microglia because the RTL9CmC protein was detected in similar granules in the isolated microglia cells cultured from P0 neonates (Figure 2c and Appendix A) [36]. In addition, we identified the RTL9-expressing lysosome granules using the lysosome-specific marker, LysoTracker Red NDN99 (Figure 2d). Thus, it turned out that the RTL9CmC protein localizes in the microglial lysosomes in the brain.

### 2.3. Involvement of RTL9 in Brain Innate Immunity against Fungi

Upon fungal zymosan injection to the brain, RTL9 colocalizes with the incorporated zymosan in microglial lysosomes and promotes its degradation. We reasoned that RTL9 must be functional in brain innate immunity like RTL5 and RTL6 because it also exhibits microglial expression. The RTL5 and RTL6 proteins in the secretory granules of microglia are secreted into the extracellular space and there await their targets, such as dsRNA and LPS, while RTL9 is specific to microglial lysosomes and does not exit into the extracellular space, implying that RTL9 reacts to pathogens that are removed via different mechanisms. We analyzed the response of RTL9 to fungal zymosan because previous reports have shown that zymosan is promptly incorporated into lysosomes [32]. Zymosan is initially recognized by cell surface receptors, such as the TLR2/TLR6 heterodimer and/or other components, then trapped in phagosomes in an LC3-dependent manner and finally degraded in phagolysosomes by fusion with lysosomes [30,37,38].

Zymosan is a complex biomolecule containing protein–carbohydrate complex elements such as glucan, mannan, chitin and protein in addition to glycolipid, and therefore exhibits specific autofluorescence patterns (shown in artificial green in Figure 3) that can be distinguished from other autofluorescence patterns in the brain using the LSM880 Automatic Component Extraction (ACE) function (Appendix A), as described in our previous report [21]. Large aggregates of zymosan are also evident in the transmission image (arrows in Appendix A), so the reliability of its autofluorescence pattern may be verified.

At 20 to 30 min after the zymosan injection into the brain, we confirmed that zymosan had colocalized with RTL9CmC (Figure 3a,b). The signal from zymosan was greatly reduced 60 min after the injection, indicating that zymosan had been degraded in the lysosomes (Figure 3c). We then examined the reaction in the *Rtl9* KO brain and found that the signal intensity remained unchanged even after 72 min (Figure 3d). This clearly demonstrates that RTL9 is essential for the zymosan degradation reaction in the microglial lysosomes, although it remains unknown how RTL9 is involved and/or what the precise role of RTL9 is in this reaction. We also found that zymosan was normally incorporated into the lysosomes even in the *Rtl9* KO brain (Figure 3e), suggesting that RTL9 may not be necessary for the fusion between the zymosan-containing phagosomes and lysosomes. We also confirmed that zymosan was normally degraded in the *Rtl5* and *Rtl6* DKO brain [25], indicating that the zymosan degradation activity is specific to *Rtl9* (Figure 3f).

## 3. Discussion

As most fungi are ubiquitous in the environment, human and animal immune systems have co-evolved and adapted to their presence over millions of years [27]. Fungal infections are typically rare yet uniquely dangerous because they are difficult to diagnose and treat. Fungal diseases cause significant morbidity and mortality, particularly in immunocompromised individuals; certain fungi can cause cutaneous lesions, acute self-limiting pulmonary manifestations, inflammatory diseases and severe life-threating infections [25,26,27,28,29,30,31]. This is particularly the case when the fungus invades the brain and central nervous system (CNS), such as fungal meningitis in humans. Many fungi in our environment can cause meningitis, including *Aspergillus*, *Blastomyces, Candida*, *Cladophialophora, Coccidioides, Cryptococcus* and *Histoplasma*. CNS fungal infections in small animals, such as cats and dogs, also cause multifocal meningoencephalomyelitis, intracranial lesions that accompany sinonasal lesions, ventriculitis, or a solitary brain or spinal cord granuloma [39]. Therefore, innate antifungal immunity is as much an important survival mechanism of organisms as those against bacteria and viruses. *RTL9* is a promising antifungal therapy target as a newly identified member of innate antifungal immunity in eutherians. Furthermore, fungal infections in patients with Alzheimer’s disease (AD) have been reported [40,41,42], suggesting that fungal infection may also be involved in the etiology of AD and/or other neurodegenerative diseases and non-infectious autoimmune diseases, such as multiple sclerosis, neuromyelitis optica, amyotropic lateral sclerosis and Parkinson’s disease [40,41,42,43,44,45,46,47]. Recent studies on innate and adaptive immunity have reported that phagocytic cells are essential players in protecting against fungal diseases and that defects in these cells compromise the host’s ability to resist fungal infection [30,31].

In this investigation, we have demonstrated that retrovirus-derived *Rtl9* is another microglial gene playing an important role in anti-fungal protection via its ability to degrade zymosan in lysosomes. The *Rtl9* mRNA level is quite low and requires 32 cycles of PCR for its detection even in the brain (Figure 1b); therefore, KI mice expressing the RTL9-CmCherry fusion protein from an endogenous locus (Figure 1c and Appendix A) were essential for the detection of the RTL9 protein in vivo, the determination of its location in the microglial lysosomes, and its involvement in the zymosan reaction (Figure 2 and Figure 3a,b). Finally, in *Rtl9* KO mice it was demonstrated that RTL9 is essential for zymosan degradation in the microglial lysosomes (Figure 3c,d). We also confirmed that the zymosan degradation activity is specific to RTL9 and not RTL5 or RTL6, other microglial proteins for the removal of dsRNA and LPS, respectively [21] (Figure 3f). It is likely that this is the reason why *RTL9* is so robustly well conserved across all eutherian species (dN/dS < 1, Table 1), because protection against fungal infection in the brain provides an evident evolutionary advantage to go along with the protection afforded against viral and bacterial infection by *RTL5* and *RTL6*, respectively [21]. Considering that *RTL9* is in charge of the eutherian-specific zymosan degradation mechanism, it may well be that *RTL9* is involved in the defense against fungal diseases in the brain.

In contrast to the widely conserved TLR system in the animal kingdom [24], that regulates cytokine and interferon production against various pathogens in a pathogen-dependent manner via each differentiated TLR protein [26,27], RTL9, RTL5 and RTL6 are apparently novel constituents in the brain’s innate immune system in eutherians [14,21]. Together with the previous study, this work demonstrates that at least three RTL genes are domesticated (exapted) in the current eutherian brain innate immune system and play important roles in the prompt removal of bacteria, viruses and fungi in a pathogen-dependent manner. However, the removal of such pathogens is a common activity of microglia/macrophages and is not specific to eutherians, suggesting the existence of alternative and/or redundant pathways in non-eutherian organisms and eutherians as well. Therefore, it is also of interest that other retrovirus-derived genes have also been exapted in non-eutherian organisms.

What is the relationship between the TLR system and RTL genes? In the case of zymosan, it is known that the TLR2/TLR6 heterodimer first recognizes zymosan on the cytoplasmic membrane, incorporates it into phagosomes associated with CD14 and then, together with alternative pathways, sends the signal downstream for gene regulation in the course of these processes [32,33]. As RTL9 apparently reacts to zymosan subsequent to fusion of the zymosan-containing phagocytes and lysosomes where RTL9 is localized, it is reasonable to think that the TLR2/TLR6 system is able to function independently of RTL9 and regulate inflammation via cytokine and interferon induction on its own. It was recently suggested that the TLR2/TLR5 heterodimer also induces inflammation when stimulated with zymosan, although TLR5 is known to primarily react to bacterial flagellin [48]. Our preliminary experiments confirmed the upregulation of *Tnfa*, *Il6* and *Il8* mRNA [49,50] in the *Rtl9* KO microglia in the same manner as control WT microglia, supporting this idea (Appendix A). It is possible that *Rtl9* itself exhibits upregulation upon zymosan administration, due to the presence of a feedback mechanism. Therefore, more detailed analysis is required to properly elucidate the relationship between the TLR system and *RTL9*, as well as the *RTL9* self-regulation mechanism.

What is the role of RTL9 in the zymosan degrading activity? Does it act as an enzyme, or a modulator? Lysosomes have a variety of hydrolytic enzymes, including more than 50 glucosidases [51]. Is RTL9 another lysosomal enzyme? Unfortunately, at present it is not possible to determine whether it has such enzymatic activity due to the lack of adequate information. 

The retrotransposon/retroviral GAG protein is comprised of three parts, the matrix, the capsid and the nucleocapsid, and each part is generated by the POL proteases. They are structural proteins for virus formation, and therefore have no intrinsic enzymatic activities. Recently, certain GAG-derived genes, such as *ARC*, *PEG10* and *RTL1*, as well as other RTL and Paraneoplastic Ma antigen (PNMA) genes, have attracted attention because they form virus-like particles that are able to mediate cell–cell communication by sending mRNA/siRNAs, and *PEG10* in particular instances, as a delivery vehicle for specific mRNAs [52]. RTL5 and RTL6 are secreted proteins that trap dsRNA and LPS, respectively, in the brain [21], implying that they play a role as structural proteins. Therefore, we reason that the GAG-like region of RTL9 forms a complex with zymosan to promote (or modulate) the degradation of this complex by an oxidative mechanism. This is because that zymosan is easily solubilized to a high molecular weight polysaccharide via the activity of the hypochlorous acid generated by the heme protein myeloperoxidase (MPO), suggesting that oxidative degradation is the main metabolic pathway of particulate (zymosan) β-glucans [53].

BLLF1 comprises 907 aa and is a type 1 membrane protein also known as glycoprotein 350 (gp350). The large N-terminal segment (1–860 aa) extends outside the viral membrane, followed by the transmembrane/cytoplasmic tail regions (861–907 aa) [54]. It is known that its extreme N-terminus portion (1–470 aa, domains I, II and III) can bind to CR2, the receptor of Epstein–Barr virus, but it is not essential for this binding reaction because of the presence of alternative proteins that fulfill the same function [55]. It is structurally well-defined [56], but the RTL9 homologous portion (473–780 aa) just after it is functionally and structurally undetermined [54] (Appendix A). 

UL36 consists of 3,164 aa and possesses two leucine-zipper motifs, a proline-glutamine (PQ) repeat, and several ATP-binding sites [57]. Its RTL9 homologous portion (787–1141 aa) corresponds to the PQ repeat region, but there are no apparent PQ repeats in RTL9 (Appendix A). Although there are several PQ motifs throughout the entire RTL9 protein, no PQ motifs are conserved in ten of the eutherian species (six PQ motifs are conserved in at least eight species) (Appendix A, green dashed boxes). It is reported that the PQ motif on a loop in the cystinosin protein, the lysosomal cystine transporter, is critical for cystinosin transport to lysosomes [58,59]. It is therefore possible that certain PQ motif(s) play a role in RTL9 transport to lysosomes. Unfortunately, there is no RTL9 3D structure prediction afforded by Alphafold 2, except the GAG-like region. Therefore, further studies are obviously required to elucidate the biochemical role of RTL9 in the zymosan degradation reaction in detail.

In this study, the importance of the RTL9 C-terminal part in zymosan degradation has been demonstrated. This activity potentially arises out of the entire sushi-ichi GAG (the maximal GAG-like region) or possibly comprises the latter half of UL36-like region and the minimal GAG-like region (Figure 1a and Appendix A). However, it is also possible that the remaining large regions contribute to the enzymatic zymosan degrading activity in combination with the C-terminal region. We confirmed that the incorporated zymosan ultimately reached the microglial lysosomes even in *Rtl9* KO mice, meaning that the C-terminal portion is not essential for either zymosan incorporation into phagosomes or the phagolysosome fusion process (Figure 3d). However, it is also possible that the other regions may be required for these processes. Zymosan degradation is unique because it is carried out via LC3-mediated phagocytosis in association with several autophagy components [30,37,38]. Further studies are required to elucidate the entire function(s) of the RTL9 protein during these processes, including both the fusion of the zymosan-incorporated phagosomes with lysosomes and zymosan degradation by generating other KO mice in which the entire RTL9 sequence is deleted.

Gould and his colleague proposed “the concept of exaptation”, that is, gaining new function(s) for specific purposes during the course of evolution, and predicted it as one of the central mechanisms in biological evolution [60,61]. Among the 11 RTL genes, *PEG10*, *RTL1* (aka *PEG11*) and *LDOC1* (aka *SIRH7* and *RTL7*) play essential but different roles in the placenta [2,12,13,15,18,62], while *RTL5*, *RTL6* [21] and *RTL9* (this study) play important but different roles in microglia. Thus, the retrovirus-derived RTL genes are very good examples of exaptation. Microglia originate in early development from the yolk sac and not from the bone marrow, as do most macrophages [63,64]. Therefore, *RTL9* provides considerable support for a central hypothesis in our work, which is that extraembryonic tissues, such as the yolk sac and placenta, have served as incubation sites for the birth of newly acquired genes from retrotransposons/retroviruses in the course of eutherian evolution [14,21].

## 4. Materials and Methods

### 4.1. Mice

All animal experiments using *Rtl9-mCherry* KI and *Rtl9* KO mice were reviewed and approved by the Institutional Animal Care and Use Committee of Tokai University (Isehara-234011) and Tokyo Medical and Dental University (TMDU) (A2022-057A) and were performed in accordance with the Guideline for the Care and Use of Laboratory Animals of Tokai University and TMDU.

### 4.2. Comparative Genome Analysis

The sushi-ichi GAG (AAC33525.1) and mouse RTL9 (NP_001035524.2) protein sequences were obtained from NCBI (accessed on 18 April 2023) [65], while the amino acid identity and similarity were calculated using the EMBOSS water program [66] and EMBOSS needle program (accessed on 18 April 2023) [67] in the default mode. The orthologues of RTL9 were identified by a search of the NCBI Gene database [65] using RTL9 (and RGAG1) as the query terms. The RTL9 coding sequences used for the pairwise dN/dS analysis (Table 1) and the amino acid alignment (Appendix A) were as follows: Mouse (*Mus musculus*): NM_001040434.2; Rat (*Rattus norvegicus*): XM_003752123.5; Human (*Homo sapiens*): NM_001385449.1; Chimpanzee (*Pan troglodytes*): XM_016942939.3; Dog (*Canis lupus familiaris*): XM_038449037.1; Horse (*Equus caballus*): XM_005614417.3; Elephant (*Loxodonta Africana*): XM_003414859.3; Manatee (*Trichechus manatus latirostris*): XM_023734430.1; Armadillo (*Dasypus novemcinctus*): XM_023582615.1; Sloth (*Choloepus didactylus*): XM_037822012.1. The RTL9 coding sequences were translated to protein sequences using the EMBOSS Transeq (accessed on 18 April 2023) [68] program. The amino acid alignment in the ten eutherian species was constructed using the Clustal Omega program (accessed on 18 April 2023) [69] in the default mode. The conserved domains in RTL9 were identified by means of the NCBI Conserved Domains Database (CDD) (accessed on 25 October 2022) [70].

### 4.3. Estimation of the Pairwise dN/dS Ratio

The nonsynonymous/synonymous substitution rate ratio (dN/dS) was estimated with CodeML (runmode: −2) in PAML [71]. An amino acid sequence phylogenic tree was constructed with MEGA7 [72] using the Maximum Likelihood method with the JTT matrix-based model. The codon alignment of cDNA was created with the PAL2NAL program [73]. The RTL9 coding sequences used for the pairwise dN/dS ratio analysis (Table 1) are described above.

### 4.4. RT-PCR

Total RNA samples were prepared from adult tissues (11 w) using ISOGEN (Nippon Gene, Chiyoda, Japan). The cDNA was synthesized from 1 μg of total RNA using Superscript III reverse transcriptase (Invitrogen, Carlsbad, CA, USA) with an oligo-dT primer. For RT-PCR, 10 ng cDNA in a 25 μL reaction mixture containing 1× ExTaq buffer (TaKaRa, Kusatsu, Japan), 200 μM each of NTP and 800 nM of primers along with 0.625 units *ExTaq* HS (TaKaRa) were subjected to 32 cycles at 96 °C for 15 s, 65 °C for 30 s and 72 °C for 30 s in a GeneAmp PCR System 2400 (Perkin-Elmer, Waltham, MA, USA). The primer sequences used were as follows: *β*-actin-forward 5’-AAGTGTGACGTTGACATCC-3’ and *β*-actin-reverse 5’-GATCCAACTCTGCTGGAAGG-3’; *Rtl9*-forward 5’-TCACCTACATGCCTGTGACC-3’ and *Rtl9*-reverse 5’-CAACAACACCACATTGTTACGG-3’.

### 4.5. Generation of the Rtl9-mCherry Knock-in Mice

*Rtl9*-*mCherry* knock-in mice were generated by pronuclear microinjection of the CRISPR/Cas system (see Figure 1c), essentially as described in a previous report [74], using a plasmid targeting vector. The crRNA was designed and synthesized with the target sequence: 5’-TACAAACAAGTAGTACTCCT-3’ (Fasmac, Atsugi, Japan). The plasmid for mCherry insertion was constructed with 1.5 kb long 5’ and 3’ arms amplified from the C57BL/6N genome using PrimeSTAR GXL DNA Polymerase (TaKaRa). The 5’ arm is the genomic sequence upstream of the stop codon of *Rtl9* and the 3’ arm is downstream of the predictive Cas9 cut site. The C-terminus of *Rtl9* was fused to a 4× GGS linker (ggaggatcaggaggatcaggaggatcaggaggatca)-attached mCherry by means of a cloning enzyme (In-Fusion^®^ HD Cloning Kit, Takara Bio, Kusatsu, Japan). The purified targeting vector was assessed for its quality using Sanger sequencing and injected into mouse pronuclei at the final concentration of 10 ng/μL. Just before injection, the targeting vector was mixed with other CRISPR/Cas system components, including the Cas9 protein (a final concentration of 30 ng/μL), crRNA (8.7 ng/μL), tracrRNA (14.3 ng/μL) and injected into the pronuclei of mouse zygotes produced through in vitro fertilization using C57BL/6N mice. Zygotes that survived the microinjection procedure were cultured in KSOM at 37 °C under 5% CO_2_ in air and transferred into the ampulla of the oviduct of pseudopregnant ICR females. Genomic modification of the mutant mice was confirmed using 3 sets of PCR primers (5’- GGAATGATGTCCACGCCACTA-3’ and 5’- CTTCAGCTTCAGCCTCTGCT-3’, 5’-CCTGTCCCCTCAGTTCATGT-3’ and 5’-CCTAGACTATTGGACCAGAGG-3’, and 5’-GGAATGATGTCCACGCCACTA-3’ and 5’-CCTAGACTATTGGACCAGAGG-3’).

### 4.6. Generation of the Rtl9 Knock-Out Mice

*Rtl9* knock-out mice were generated using a pair of crRNAs and a single-stranded oligo donor DNA (ssODN). To cut off the C-terminal genomic region of *Rtl9*, a pair of crRNAs targeting 5’-TCATAGCTGCAAATTGTGCA-3’ and 5’-AGAAATCACATAAGATTCCA-3’ were synthesized (Fasmac). The 162-base ssODN was composed of a 3x stop codon (3× stop) and 5’ and 3’ homology arms having 75 bases each (Hokkaido System Science, Sapporo, Japan). Just before injection, CRISPR/Cas system components including the Cas9 protein (a final concentration of 30 ng/μL), a pair of crRNAs (8.7 ng/μL), tracrRNA (14.3 ng/μL) and ssODN (15 ng/μL) were mixed and injected into the pronuclei of mouse zygotes produced through in vitro fertilization using C57BL/6N mice. The zygotes that survived the microinjection procedure were cultured in KSOM at 37 °C under 5% CO_2_ in air and transferred into the ampulla of the oviduct of pseudopregnant ICR females. Genomic modification of the mutant mice was confirmed using a set of PCR primers (5’-GAGAACACCAGCTTCTAGAGC-3’ and 5’-GGGAGTTCAGAACCTCATACAC-3’).

### 4.7. Imaging Using Confocal Laser Scanning Fluorescence Microscopy

Fresh brain hemispheres from *Rtl9*-*mCherry* KI mice were used for analysis with a ZEISS LSM880 (ZEISS, Jena, Germany) without fixation. The samples were covered with 10% glycerol solution for protection from drying. The samples were observed using a Plan-Apochromat lens (10×, numerical aperture = 0.45, M27, ZEISS) and a C-Apochromat lens (63× numerical aperture =1.2 Water, ZEISS). The tiling with lambda-mode images was obtained using the following settings: pixel dwell: 1.54 μs; average: line 4; master gain: ChS: 1250, ChD: 542; pinhole size: 33 µm; filter 500–696 nm; beam splitter: MBS 458/514; lasers: 514 nm (Argon 514), 0.90%. For the tiling-scan observations, the tiling images were captured as tiles: 84, overlap in percentage: 10.0, tiling mode: rectangular grid, in size: x; 15,442.39 µm, y; 9065.95 µm. The spectral unmixing and processing of the obtained images were conducted using the ZEN imaging software (version 2.3 SP1, Carl Zeiss Microscopy, Jena, Germany). The spectrum from the mCherry protein (maximum peak emission fluorescence wavelength: 610 nm) detected in *Rtl9-mCherry* KI brain was distinguished from Af610 nm using the peak shape, such as the width and/or co-existence of a second peak. The relative intensity of RTL9-mCherry (red), zymosan autofluorescence (green) and LysoTracker Red NDN99 (red in Figure 3e or green in Figure 2d) signals along the *x*-axis of the brain (from the olfactory bulb to the cerebellum region) was calculated from 3D scanning data. The total intensity of each signal on and above each *y*-axis was summed, divided by the transmission signal and presented as the relative signal intensity on the *y*-axis in this figure.

### 4.8. Zymosan Injection to the Brain

*Rtl9*-*mCherry* KI, *Rtl9* KO and *Rtl5* and *Rtl6* double KO mice [25] (P5 to 3 w of age) were used for the injection experiments after being anesthetized with isoflurane. Ten µL of 10 ng/mL solution of zymosan (NOVUS BIOLOGICALS, Centennial, CO, USA, NPB2-26233, 1/5000 dilution) was injected using 1 mL insulin syringes and a 26 G needle. Then, 1 min after the injection, the needle was pulled out and kept out for 5 min; then, the fresh brain was dissected out in cold PBS solution. The inner surface of the brain hemispheres was analyzed with a ZEISS LSM880 (ZEISS, Jena, Germany).

### 4.9. Lysosome Staining with LysoTracker

A 1/50 dilution of LysoTrackerTM Red DND-99 (ThermoFisher Scientific, Waltham, MA, USA, L7528, 1 mM) was used for the lysosome staining of the fresh brains. A 100 µL solution was placed beneath the brain hemispheres and reacted for 30 min at RT. The LysoTracker signal (emission peak at 590 nm) was separated from various kinds of autofluorescence (Af) in the brain using the LSM880 ACE function.

## 5. Conclusions

We have demonstrated that eutherian-specific *RTL9* has functions in the innate antifungal immunity in the brain because *Rtl9* KO mice lost zymosan degradation activity. Fungi are one of the most dangerous infectious pathogens, along with viruses, bacteria and protozoa. This work provides compelling evidence that *RTL9*, along with *RTL5* and *RTL6*, plays an important role in the brain’s innate immunity, suggesting that certain retrovirus-derived genes contributed to the evolution of the brain’s innate immunity in a eutherian-specific manner. Phagocytic cells, such as microglia/macrophages, are essential for the defense against fungal infection. As defects in these cells reduce host resistance to fungal infection, *RTL9* is a novel target for antifungal therapy as well as neurodegenerative diseases as a newly identified member of innate antifungal immunity in eutherians.

## Figures and Tables

**Figure 1 ijms-24-14884-f001:**
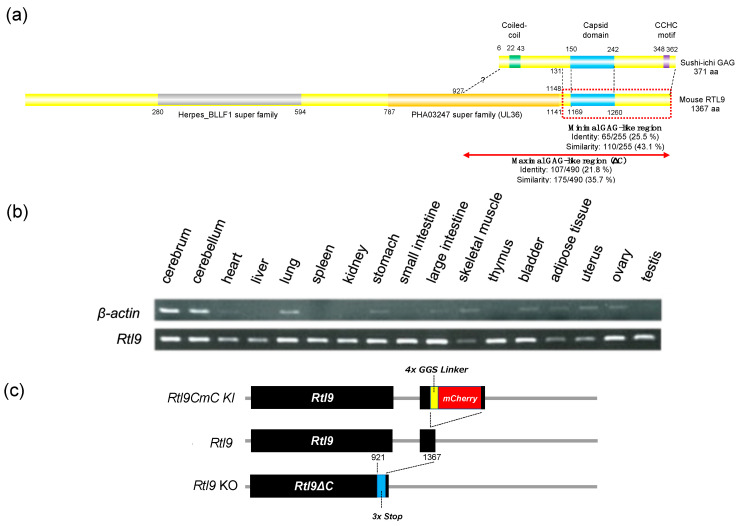
Features of the RTL9 protein, mRNA expression, and gene structures of KI, normal and KO mice. (**a**) Alignment of the sushi-ichi GAG and mouse RTL9 protein. Coiled-coil motifs in the N-terminus, a capsid domain in the middle and a zinc finger CCHC domain in the C-terminus are represented as green, blue and purple boxes, respectively. The Herpes BLLF1 super family and PHA03247 super family (UL36) motifs are depicted with a gray and orange box, respectively, and the capsid domain of the sushi-ichi retrotransposon is presented as a blue box. Minimal and maximal Gag-like regions are presented with a red dashed line and indicated by a red double-headed arrow, respectively. (**b**) Mouse *Rtl9* mRNA expression in the adult tissues and organs at 19 weeks. The RT-PCR products use total RNA (10 μg, 32 cycles for *Rtl9* and 26 cycles for *β-actin* as a control). (**c**) The generation of the *Rtl9CmC* KI and *Rtl9* KO mice. Top: the open frame of the mCherry protein (red) is fused with the C-terminus of RTL9 via a 4 x GGS linker (yellow). Middle: mouse *Rtl9*.

**Figure 2 ijms-24-14884-f002:**
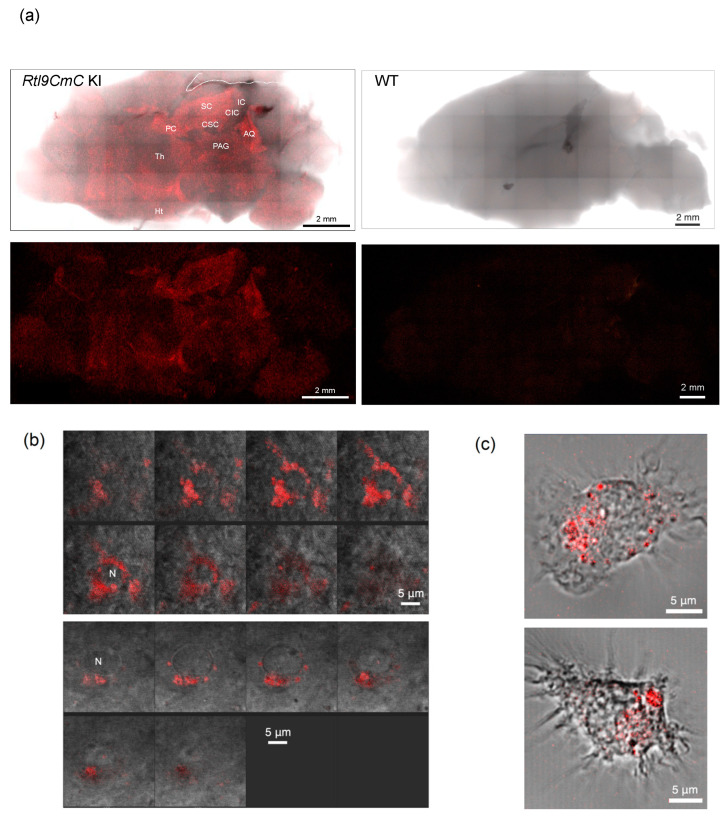
Expression of the RTL9CmC protein in the brain and isolated microglia. (**a**) A high expression of RTL9-CmC (red) was detected in the cerebral aqueduct (AQ), commissure of the superior and inferior colliculus (CSC and CIC), periaqueductal gray (PAG), posterior commissure (PC) and superior colliculus (SC), but not in the inferior colliculus (IC), hypothalamus (Ht) or thalamus (Th). The mCherry signals were almost negligible in the WT control. Left: *Rtl9CmC* KI brain (P15). Right: WT control brain (P18). *n* > 8. (**b**) Two sequences of photographs at a 0.5 μm interval indicated that the RTL9CmC protein accumulated in small granules around nuclei in the round cells in the brain (P5). *n* = 5. (**c**) The RTL9CmC protein was detected in both small and large granules in isolated microglia cultured from the P0 brain. *n* = 2. (**d**) The RTL9CmC protein (red) was colocalized with LysoTracker (artificial green), a specific lysosome marker, in the P9 brain. *n* = 3. N: nuclei.

**Figure 3 ijms-24-14884-f003:**
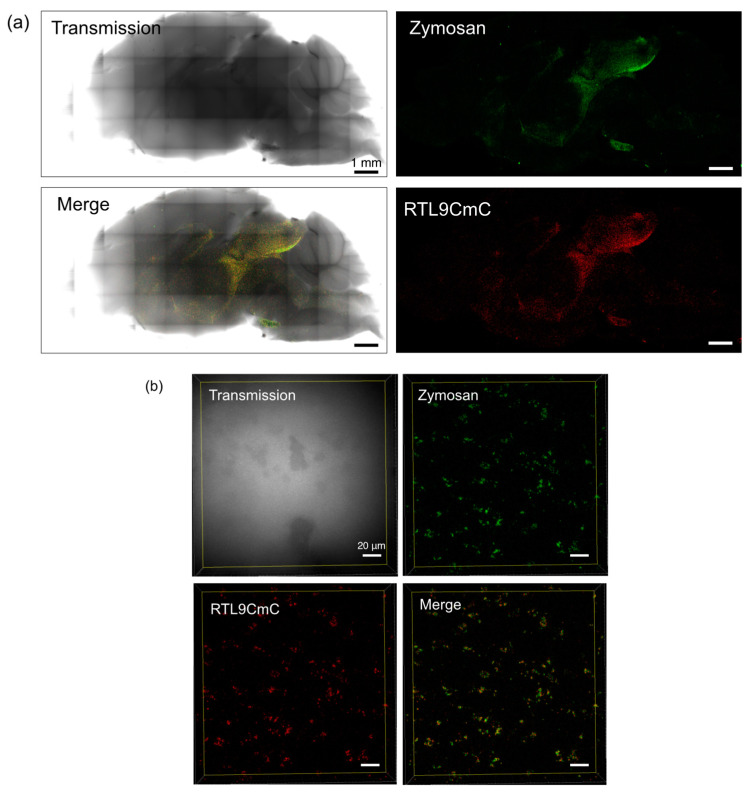
Reaction between the RTL9CmC protein and zymosan. (**a**) Colocalization of RTL9CmC (red) and zymosan (green) in the brain. *n* = 5. (**b**) A higher magnification image of the midbrain region. *n* = 3. (**c**,**d**) Time course of zymosan degradation in *Rtl9CmC* KI ((**c**) 20 and 60 min) and *Rtl9* KO brain ((**d**) 20 and 72 min), respectively. *n* = 2. (**e**) Lysosomal localization of injected zymosan in *Rtl9* KO brain: 90 min after injection, Lysotracker was added and reacted for 30 min at RT, then a fluorescent image was obtained. *n* = 3. (**f**) Normal zymosan degrading activity in *Rtl5/Rtl6* DKO brain. *n* = 3.

**Table 1 ijms-24-14884-t001:** Conservation of *RTL9* in eutherian species. Pairwise dN/dS ratio of RTL9 in 10 representative eutherian species was shown. The human, chimpanzee, mouse, rat (Euarchontoglires), dog and horse (Laurasiatheria), manatee and elephant (Afrotheria), and armadillo and sloth (Xenarthra) species represent all four major groups of eutherians.

dN/dS	Human	Chimpanzee	Mouse	Rat	Dog	Horse	Manatee	Elephant	Armadillo
Chimpanzee	0.7009								
Mouse	0.4544	0.4529							
Rat	0.4022	0.4089	0.3364						
Dog	0.4272	0.4408	0.4236	0.3936					
Horse	0.6116	0.6153	0.4687	0.4153	0.4181				
Manatee	0.5264	0.5334	0.4170	0.4060	0.4148	0.5173			
Elephant	0.5409	0.5488	0.4181	0.4178	0.4751	0.5973	0.4284		
Armadillo	0.5552	0.5638	0.5075	0.4542	0.4794	0.5588	0.5227	0.5334	
Sloth	0.6619	0.6803	0.5726	0.5493	0.5014	0.6432	0.6154	0.6365	0.7614

Mouse: house mouse, rat: Norway rat, manatee: Florida manatee, elephant: African savanna elephant, armadillo: nine-banded armadillo, sloth: Hoffmann’s two-fingered sloth.

## Data Availability

Data can be made available upon request from the corresponding author.

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
