# Peer review of "Retrovirus-Derived RTL9 Plays an Important Role in Innate Antifungal Immunity in the Eutherian Brain"

_ijms, 2023, doi:10.3390/ijms241914884_

Round 1

Reviewer 1 Report

The manuscript by Isjino et al, titled "Retrovirus-derived RTL9 plays an important role in innate antifungal immunity in the eutherian brain provides valuable information about antifungal immunity mediated by the RTL9 protein. I believe the manuscript is technically sound and contains novel information for future readers.

I have a few minor and one major comment for the authors to address.

Minor comments:

Line 29, rtl9 is written as small alphabets, kindly check. 

Line 76 add space between pH and the range.

In Line 132, it should be "μg".

Line 133, add space between "the" and "Rtl9".

Major comment

Kindly check Figure 1, in the legend it is written that "Alignment of the sushi-ichi GAG and mouse RTL9 protein. Coiled-coil motifs in the N-terminus, a capsid domain in the middle, and a zinc finger CCHC domain in the C-terminus are represented as green, blue, and purple boxes, respectively", but I do not see any colors in Figure 1. I think the Figure 1 has to be done again. Also, in Figure 1B, the text for the names of the organs seems strange---kindly check.

Author Response

Thank you very much for the favorable comments from the reviewer 1.

All points raised were correctly addressed as follows.

Minor comments:

Line 29, rtl9 is written as small alphabets, kindly check.

Thank you for your comment. In this manuscript, we always use Rtl9 (italic) for mouse gene, while RTL9 (italic) is used for humans and other animals. RTL9 protein is always used without exception.

Line 76 add space between pH and the range.

In Line 132, it should be "μg".

Line 133, add space between "the" and "Rtl9".

Thank you for your indications. We amended them accordingly.

Major comment

Kindly check Figure 1, in the legend it is written that "Alignment of the sushi-ichi GAG and mouse RTL9 protein. Coiled-coil motifs in the N-terminus, a capsid domain in the middle, and a zinc finger CCHC domain in the C-terminus are represented as green, blue, and purple boxes, respectively", but I do not see any colors in Figure 1. I think the Figure 1 has to be done again.

I think something went wrong with this figure in your file. We have checked the Figure 1 is correctly colored. Could you check our revised manuscript, again?

Also, in Figure 1B, the text for the names of the organs seems strange---kindly check.

Thank you for your points. We correctly amended them, such as small intestine, large intestine and adipose tissue.

Reviewer 2 Report

The interesting manuscript by Dr. Fumitoshi Ishino and colleagues entitled " Retrovirus-derived RTL9 plays an important role in innate antifungal immunity in the eutherian brain" which expands the poor knowledge about the brain defense of some mammals, including humans, against fungal infections, meets the requirements for manuscripts submitted to the editorial office of International Journal of Molecular Sciences (IJMS), unfortunately, it concerns animal research and the authors have made declarations that seem to only authorize them to publish the results only in a regional University professional journal addressed to a narrow group of specialists. After obtaining consent from the Independent Commission for research on animals in the topic described by the authors, providing the name of the commission and the consent number, extending the introduction (and/or discussion) with several current literature reports and removing several minor shortcomings listed below, this manuscript will be able to be Successfully recommended for subsequent stages of production in the IJMS Journal.

·        Please update the source and review literature.   ·        Websites should be cited in the usual way, including them in the "References" section, including the date of visit, as the content of websites is usually updated, and please keep the content referred to by the authors in case readers ask in the future.

·        Please add two subsections, "Summary" and "Abbreviations". The Abstract is independent of the entire manuscript, therefore a Summary is needed and the accumulation of abbreviations requires their alphabetical listing with an explanation, which will make reading easier for the reader who specializes in related fields.

·        This interesting manuscript contains only 10 references to current source literature. Therefore, please expand the introduction with more current literature, even from fields closely related to the discussed topic.

·        On lines 4–5 is: … Fumitoshi Ishino1*#, Johbu Itoh2%, Masahito Irie1, 3$, Ayumi Matsuzawa1, 4&, Mie Naruse1@, Toru Suzuki5, Yuichi Hiraoka5, 6, and Tomoko Kaneko-Ishino3* … , but maybe should be Fumitoshi Ishino 1*, Johbu Itoh 2, Masahito Irie 1, 3, Ayumi Matsuzawa 1, 4, Mie Naruse 1, Toru Suzuki 5, Yuichi Hiraoka 5, 6, and Tomoko Kaneko-Ishino 3* … . Comment: Please check the recommendations for authors and correct them as there are unauthorized characters: " # ", " % ", " $ ", and " & " twice.

·        On line 21 there is: … knockin … , but maybe should be … knocking … . Comment: Please add “ g “, if necessary.

·        On line 24 is: … Futhermore, … , but should be … Furthermore, … . Comment: Linguistic mistake – please add “ r “.

·        On line 37 there is: … [1-5] … , but rather it should be … [1–5] … . Comment: Recently, the medium " – " sign between numbers has been replaced by the short " - " sign. Similar minor errors to correct are found in the lines: 39, 43, 44, 50, 68, 76, 98, 100, 105, 106, 111, 123, 125, 141 (twice), 214, 342, 243, 344, 347, 351, 372, 550, 553, 556, 558, 576, 580, 584, 586, 596, 598, 601, 604, 610, 618, 619, 621, 624, 625, 628, 629, 631, 633, 635, 647, 651, 655, 658, 664, 668, 670, 672, 675, 678, 683, 685, and 687. See lines 482, 570, 573, 582, 593, 613, 639, 643, 660, and 666.

·        On line 223 please decode the sentence … Twenty-30 min … . Comment: Although the text is correct, it may be considered an error by readers. Please use a descriptive style with only numbers, alternatively, describe the time period only in words.

·        On lines 298, and 304 are: … exapted … , but please check and correct the linguistic mistake.

·        On line 300 is: … manner. However … , but should be … manner. However … . Comment: The bold dot and the space after the dot should be written in normal style.

·        On lines 453, and 470 are: … CO2 … , but should be … CO2 … . Comment: In summary chemical formulas, in accordance with the recommendations of the IUPAC nomenclature, the number of atoms of a given element should be written in a subscript.

·        On line 647 (ref. [40]) after the year, please insert a comma “ , “.

·        On line 664 (ref. [47]) is: … 74,10142-10152 … , but should be … 74, 10142–10152 … . Comment: The comma “ , “ should rather be written in a normal font “ , “, and please insert a space character after the comma.

On line 675 (ref. [53]), please write the year in bold.

Author Response

Please see an attached file

Round 2

Reviewer 2 Report

Significant shortcomings of the manuscript entitled "Retrovirus-derived RTL9 plays an important role in innate antifungal immunity in the eutherian brain" have been corrected by the authors and in this version I recommend it for publication in the journal Int. J. Mol. Sci. after correcting the summary and removing several minor editorial errors in the cited literature listed below:

·        At line 526 is: … Conclusions … , but should be better … 5. Conclusions … . Comment: Please assign a number for the conclusions chapter.

·        At line 533–538 (Conclusion) is: … We have previously demonstrated that eutherian-specific RTL5 and RTL6 are also microglial genes that play important roles in the brain’s innate immunity against viruses and bacteria, suggesting that certain RTL genes contributed to the evolution of brain’s innate immunity in a eutherian-specific manner. … , but this sentence is inappropriate for a summary of the current manuscript. Comment: Extensive reference to the authors' previous scientific achievements should not be made in the current summary, and the summary itself could be slightly more extensive, if possible. Please revise the text.

·        The authors changed the style in the supplemented source literature. Please standardize it. Some specific notes regarding the writing style of the source literature are listed below:

At line 658 is: … E.P, Baden, L.R., … , but should be … E.P.; Baden, L.R.; … . Comment: Please synchronize the style of the cited source literature.

At line 660 is: … G.W., Ramos R.L., … , but should be G.W.; Ramos R.L.; … . Comment: Please synchronize the style of the cited source literature.

At lines 660–661 is: … system, Virulence 2017 … , but should be … system. Virulence 2017 … . Comment: Incorrectly, the name of the journal appeared at the end of the monograph's title. Please check and correct.

At line 689 is: … D., Nemergut, M., Liskova, B., … , but should be … D.; Nemergut, M.; Liskova, B.; … . Comment: Please synchronize the style of the cited source literature.

At line 690 is: … 20:25. … , but maybe should be … 20, 25. … . Comment: Please synchronize the style of the cited source literature.

At line 691 is: … D., Alonso, R., Rábano, A., Rodal, I., … , but should be … D.; Alonso, R.; Rábano, A.; Rodal, I.; … Comment: Please synchronize the style of the cited source literature.

At line 692 is: … Sci Rep, 2015, 5:15015 …, but should be … Sci. Rep. 2015, 5, 15015 … . Comment: Please synchronize the style of the cited source literature.

At line 693 is: … R., Pisa, D., Fernández-Fernández, A.M., … , but should be … R.; Pisa, D.; Fernández-Fernández, A.M.; … . Please synchronize the style of the cited source literature.

At line 694 is: … 2018, 10. … , but should be … 2018, 10, 159. … . Comment: The publication number is missing. Please check and correct.

At line 695 is: … C., Jiang, M.L., Jiang, R., Pang, T., … , but should be … C.; Jiang, M.L.; Jiang, R.; Pang, T.; … . Comment: Please synchronize the style of the cited source literature.

At line 696 is: … 13:1077335. … , but should be … 13, 1077335. … . Comment: Please check and please synchronize the style of the cited source literature.

Author Response

Thank you very much for your helpful comment on the content of "Conclusions".  We amended this part and change its position in the middle of the conclusions as below:

Fungi are........ This work provides compelling evidence that RTL9, along with RTL5 and RTL6, plays an important role in the brain innate immunity, suggesting that certain retrovirus-derived genes contributed to the evolution of brain’s innate immunity in a eutherian-specific manner. Phagocytic cells ....   We have amended the references according to the reviewer 2's indications.